# Aqueous amine enables sustainable monosaccharide, monophenol, and pyridine base coproduction in lignocellulosic biorefineries

Li Xu[1], Meifang Cao[1], Jiefeng Zhou[1], Yuxia Pang[1], Zhixian Li[1], Dongjie Yang[1], Shao-Yuan Leu[2], Hongming Lou[1] ✉, Xuejun Pan[3] & Xueqing Qiu[4] ✉

Thought-out utilization of entire lignocellulose is of great importance to achieving sustainable and cost-effective biorefineries. However, there is a trade-off between efficient carbohydrate utilization and lignin-to-chemical conversion yield. Here, we fractionate corn stover into a carbohydrate fraction with high enzymatic digestibility and reactive lignin with satisfactory catalytic depolymerization activity using a mild high-solid process with aqueous diethylamine (DEA). During the fractionation, in situ amination of lignin achieves extensive delignification, effective lignin stabilization, and dramatically reduced nonproductive adsorption of cellulase on the substrate. Furthermore, by designing a tandem fractionation-hydrogenolysis strategy, the dissolved lignin is depolymerized and aminated simultaneously to co-produce monophenolics and pyridine bases. The process represents the viable scheme of transforming real lignin into pyridine bases in high yield, resulting from the reactions between cleaved lignin side chains and amines. This work opens a promising approach to the efficient valorization of lignocellulose.

In pursuit of reducing the dependence on depleting fossil resources and meeting the rising energy requirements, it is an attractive but challenging opportunity to process inedible biomass into various biobased products and bioenergy in lignocellulosic biorefineris[1–3]. Given the structural features of lignocellulose components, achieving fermentable sugar release for biofuel production is a well-known and sought-after illustration for structural carbohydrate utilization (cellulose and hemicelluloses, accounting for 60–85 wt%)[4]. Valorizing lignin (accounting for 15–30 wt%) composed of phenylpropanoid units into useful chemicals is another key to amplifying the economic viability of future biorefineries[5].

According to the developing trend of biorefineries, the central conception is increasingly shifting from the traditional carbohydrate-first strategy toward the recent lignin-first scenario[6]. Historically, studies on lignocellulosic biorefinery focused exclusively on optimizing the utilization of carbohydrates (e.g., for cellulosic ethanol production) under harsh processing conditions[7]. Lignin was always considered a collateral product to extract some extra value. Within this context, the processability of the isolated lignin into valuable monomers was largely restricted owing to its highly condensed structures (more interlinked C–C bonds)[8,9]. Recently, recognizing the enormous potential of lignin as the most abundant aromatic reservoir in nature, a

[1]Guangdong Provincial Key Lab of Green Chemical Product Technology, State Key Laboratory of Pulp and Paper Engineering, School of Chemistry and Chemical Engineering, South China University of Technology, Guangzhou 510641, China. [2]Department of Civil and Environmental Engineering, The Hong Kong Polytechnic University, Hong Kong, China. [3]Department of Biological Systems Engineering, University of Wisconsin-Madison, Madison, WI 53706, USA. [4]School of Chemical Engineering and Light Industry, Guangdong University of Technology, Guangzhou 510006, China. ✉e-mail: cehmlou@scut.edu.cn; qxq@gdut.edu.cn

new biorefinery scheme has emerged, termed lignin-first, in which active stabilization approaches are developed to avoid condensation reactions that lead to more recalcitrant lignin structures[10]. This emerging biorefinery paradigm, which targets deriving more value from lignin than carbohydrate-first processing, is generally accomplished by the use of protection-group chemistries or reductive stabilization of reactive lignin intermediates[6]. To name a few, α, γ-diol group of β−O−4′ motif was stabilized by adding formaldehyde to form 1,3-dioxane rings[11], stable acetal was formed between diols and unstable C2-aldehyde fragments[12,13], the $C_\alpha$ alcohol of β−O−4′ was oxidized to a ketone[14], and early-stage catalytic conversion of lignin was achieved by Reductive Catalytic Fractionation (RCF) with 2-propanol as an H-donor[15,16]. Although carbohydrates could be upgraded to dissolving pulp in lignin-first biorefining[17], the enzymatic hydrolyzability of pretreated lignocellulose did not perform as well as those obtained in carbohydrate-first strategy. In other words, a high cellulase dosage (-30 FPU/g glucan[11,13,18] or more[15,19,20]) is usually required to enable efficient enzymatic carbohydrate conversion. Therefore, the development of an effective strategy to upgrade the entire lignocellulose is still in demand.

On the other aspect, although considerable research efforts have been devoted to the efficient transformation of lignin into chemicals, most products are limited to C, H, and O-containing compounds[21]. N-participated lignin conversion, which targets sustainable heteroatom-functionalized monomer production, is of great importance to expanding the product pool of lignin to meet value-added biorefining demands[22]. The N-functionalized products is seen as vital synthetic scaffolds for pharmaceuticals, agrochemicals, and polymer materials[23–25]. To date, N-containing chemicals, including pyrimidines[26], indoles[27], benzylamines[28], cyclohexylamines[29], quinolines[30], and quinoxalines[31], have been synthesized successfully from lignin model compounds, but not directly from real lignin.

In this work, we attempt to develop a mild pretreatment approach with an amine-water mixture to realize a win-win for both carbohydrate and lignin utilization, and diethylamine (DEA) is selected due to its high basicity and nucleophilicity (-$10^5$ times more nucleophilic than $NH_3$ in $H_2O$)[32,33]. This process simultaneously produces a carbohydrate fraction that is susceptible to enzymatic hydrolysis and a high-quality lignin that delivers high monomer yields upon catalytic amination and depolymerization. The N-functionalization of products is achieved in a direct manner via a consecutive fractionation-hydroprocessing step. Moreover, most of lignin depolymerization methodologies only focus on the utilization of the aromatic nuclei in lignin over the years. Present day, we put forward a biorefinery concept termed Upgradation of Lignin Side Chains, wherein the high-value conversion of cleaved lignin side chains should be taken into account, which is often overlooked in other conceptions. This biorefinery paradigm will be helpful for advancing the economic feasibility of lignin-to-chemicals valorization and pivotal for further progress in this exciting research area.

## Results

### Evaluation of pretreatment and saccharification efficiency

Corn stover (CS) was chosen as the lignocellulosic feedstock in our work. CS was pretreated at 130 °C for 1 h using aqueous diethylamine (DEA) at different concentrations. The pretreatments using ammonia (≥28% in $H_2O$) and deep eutectic solvent (DES) were performed for comparison. As shown in Fig. 1a, after water washing to remove soluble lignin from the lignocellulosic matrix (route A), two fractions, a cellulose-rich solid with high enzymatic digestibility and a lignin-rich liquid, were produced from the DEA pretreatment of CS. In this section, the pretreatment efficiency and the enzymatic hydrolyzability of the pretreated solid were evaluated.

Notably, a high biomass loading of 30 wt% was employed in the pretreatment. The high-solid loading possesses many advantages over low-solid loading (≤10 wt%), such as higher energy efficiency[34,35], lower

pretreatment reagent consumption and environmental impact[36], and less liquid waste generation. However, high solid loading limits the mass transfer in the system, often resulting in unsatisfactory and inhomogeneous pretreatment in many methods. As shown in Fig. 1c and Supplementary Fig. 1, low delignification (12.3%) and high biomass recovery (89.7%) were observed in the high-solid DES fractionation. In contrast, ammonia and amine fractionation performed much better under high-solid conditions. The lignin removal and solid recovery in ammonia pretreatment were 60.9% and 62.9%, respectively, and the cellulose content reached 59% (Supplementary Fig. 2). In DEA-based fractionation, the lignin removal was low (31.8% or 20.0%) under the condition without water (1.41 ml DEA/g, namely pure DEA) or with too much water (0.05 ml DEA/g, namely DEA:water = 5:95 v/v), while good delignification was observed at a DEA content in the range of 0.21 (DEA:water = 20:80 v/v) to 1.22 ml/g (DEA:water = 80:20 v/v). Pretreatments with 0.79 (DEA:water = 60:40 v/v) and 0.48 ml DEA/g (DEA:water = 40:60 v/v) resulted in a similar lignin removal (-74%), which outperformed ammonia and DES. It was found that the pretreatment efficiency of DEA-based fractionation relied on the presence of water. We surmised that DEA had stronger reactivity toward lignin and facilitated the dissolution of lignin out of the lignocellulose matrix more efficiently in the presence of water. The projection map of the molecular electrostatic potential in Supplementary Fig. 3 confirmed the formation of electrostatic potential penetration between electron-rich nitrogen (N) atoms and water molecules, which endowed N in DEA with stronger nucleophilicity. Additionally, water is necessary for DEA ionization to form its superior basicity. As shown in Fig. 1d, lignin was dissolved very well in the DEA-water mixture, while lignin was almost insoluble in pure DEA. It is widely known that lignin is a typical amphiphilic polymer that has both hydrophobic and hydrophilic domains. The aggregation or solvation behaviors of lignin macromolecules in solvents are largely determined by the interactions between solvent molecules and different lignin groups[37]. The radial distribution function (RDF) was employed to analyze the distribution of solvent molecules around the hydrophobic benzene ring and hydrophilic hydroxyl of lignin. The g(r) of DEA molecules to the lignin benzene ring was always larger than that of water molecules at the same distance, indicating that DEA molecules grouped around the hydrophobic functional groups of lignin more compactly and then caused the solvation of lignin hydrophobic skeletons. With respect to lignin hydroxyl, the g(r) value of water molecules was higher than that of DEA molecules at short distances (0−3 Å), suggesting that water molecules tended to be more distributed in the region around lignin hydroxyl than DEA molecules and subsequently accelerated the solvation of lignin hydrophilic fractions. Thus, we attribute the greater pretreatment efficiency of the aqueous DEA system to the stronger reactivity of DEA toward lignin and the easy solvation of lignin.

Among the pretreated CS substrates, the CS pretreated by 40% DEA delivered the highest fermentable sugar yield (85.9% glucose and 78.0% xylose, respectively) with a cellulase (CTec 2) dosage of as low as 8 FPU/g glucan, as presented in Fig. 1e. On the other aspect, as the pretreatment system remained solid-state at high solid loading, DEA-treated CS could be enzymatically hydrolyzed directly without washing and detoxification steps. As shown in Fig. 1a (route B), superb sugar yields were achieved upon direct saccharification in a substrate loading range of 4% to 8% (>87% glucose and >74% xylose, respectively). The further increase in the solid loading to 10−14% came a fall in the enzymatic hydrolysis, but glucose and xylose yields still maintained a high level of 73−77% and 63−66%, respectively. The findings verified the feasibility of one-pot saccharification of biomass by combining DEA pretreatment with enzymatic hydrolysis. Usually, cellulase inhibitors (e.g., soluble lignin-derived phenolics) generated in lignocellulose fractionation are deemed an important factor restricting the efficiency of enzymatic cellulose hydrolysis[38]. To investigate whether similar inhibitors were produced in DEA-based fractionation, the

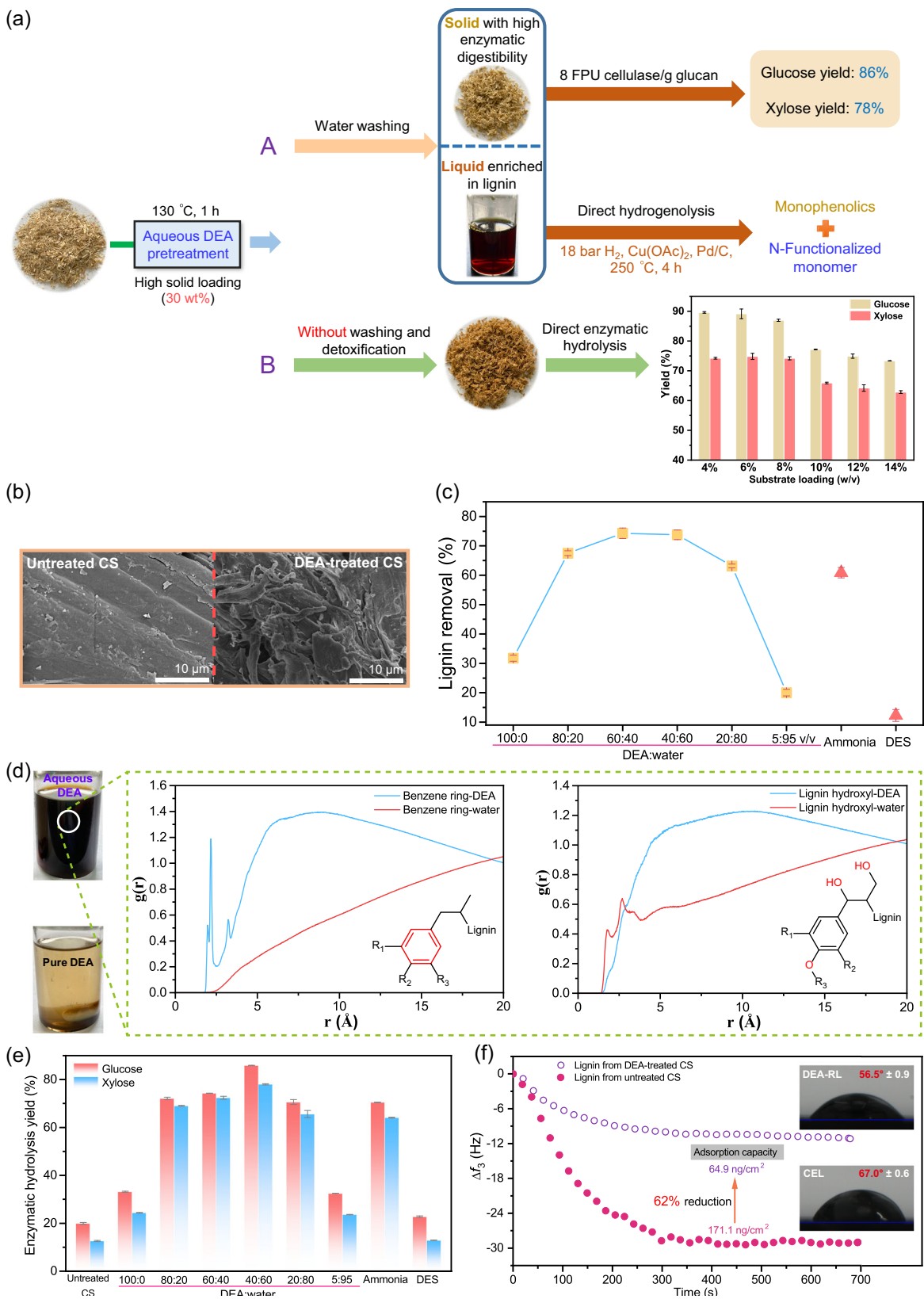

lignin-rich stream collected from the water washing of DEA-treated CS was added to the enzymatic hydrolysis system (see Supplementary Fig. 4). Interestingly, enzymatic hydrolysis performance was far from being inhibited but enhanced in the presence of the soluble matters from DEA fractionation. Taking corn cob residue as an example, a 10% v/v addition of DEA pretreatment liquor gave an optimal boost to cellulose hydrolysis. When the addition was further increased to 20–30%, enzymatic hydrolysis was still improved, albeit the promoting effect went slightly down. The results strongly suggested that no cellulase inhibitors were produced during the DEA-based fractionation. On the contrary, the lignin-rich stream promoted the enzymatic saccharification of lignocellulose.

**Fig. 1 | Development of high-solid amine-based fractionation for the covalorization of carbohydrate and lignin. a** Two biorefinery designs with DEA-based lignocellulose fractionation. A: Utilization of solid residues and lignin-rich stream after water washing step. B: Direct enzymatic hydrolysis of DEA-treated CS without any washing and detoxification. Saccharification condition: 15 FPU CTec2/g glucan, 50 °C, pH 4.8, 150 rpm, and 72 h. **b** SEM images of untreated and 40% v/v DEA-treated CS. **c** Lignin removal in different pretreatments. **d** Radial distribution function (RDF) of DEA and water molecules around hydrophobic (benzene ring) and hydrophilic (hydroxyl) functional groups of lignin in the DEA-water mixture

solvent. Insert images: photos of 40 mg of enzymatic hydrolysis lignin (EHL) in 15 ml of DEA-water (40:60 v/v) and pure DEA. **e** Glucose and xylose yield of untreated CS and CS treated with different pretreatment reagents. The enzyme dosage was 8 FPU CTec2/g glucan. **f** QCM frequency changes during the enzyme adsorption on lignin films, and comparison of water contact angle on DEA-RL and CEL surfaces. CS corn stover, DEA diethylamine, DES deep eutectic solvent, DEA-RL residual lignin isolated from 40% v/v DEA-treated CS solids, CEL cellulolytic enzyme lignin isolated from untreated CS. Error bars represent the standard deviation.

Next, we analyzed the major factors behind the increased enzymatic hydrolyzability of DEA-treated CS. The changes in biomass morphologies before and after the pretreatment were examined by SEM imaging (Fig. 1b). Compared to a flat and smooth surface of untreated CS, the surface of DEA-treated CS became rough, along with exposed slender fibers, suggesting that the ultrastructural architecture of biomass was disrupted and fragmented effectively. These morphological changes were largely due to the removal of cell wall components (mainly lignin that highly restricts the accessibility of enzymes to the cellulose surface by forming a physical barrier)[39]. There was a positive correlation between carbohydrate conversion and delignification (Supplementary Fig. 5). Although numerous pretreatments reported were also effective in lignin removal or relocation, nonproductive adsorption of cellulase onto residual lignin in pretreated substrates still exerted pronounced inhibitory effects on enzyme performance, thereby reducing sugar yields[40,41]. Quartz crystal microgravimetry (QCM) was employed to explore the cellulase adsorption dynamics on different residual lignins (Fig. 1f). The results demonstrated that irreversible adsorption of cellulase to lignin was cut by 62% after DEA pretreatment. The maximum enzyme adsorption on the lignins isolated from the CS treated with acidic ethanol/water solution, hot water, and NaOH was measured to be 224.3, 153.9, and 162.6 ng/cm², respectively, in our previous work[42], which are pronouncedly larger than that on DEA-RL (residual lignin in the DEA-treated CS). XPS spectra of N 1s in Supplementary Fig. 6 confirmed that hydrophilic N was incorporated into DEA-RL. The overall hydrophilicity of lignin was increased, which was proven by the smaller water contact angle observed in DEA-RL (see Fig. 1f). The hydrophobic interaction, which provides a long-range attraction, was proven to be the main driver for the adsorption behavior of enzymes on lignin[42]. Based on these findings, we proposed the dissimilarity in adsorption of cellulase on lignins obtained from DEA and traditional pretreatment methods (Supplementary Fig. 7). The increased hydrophilicity of DEA-RL significantly reduces its binding ability with cellulase compared to lignins from other pretreatment methods. Next to the lignin factor stated above, the enzymatic digestibility of cellulose itself in pretreated lignocellulose is also an important aspect affecting final carbohydrate conversion. As a semicrystalline natural polymer, its crystallinity index (CrI) divided by the cellulose content is regarded as a good indicator for evaluating the proportion of disordered cellulose that is susceptible to enzymatic hydrolysis[43]. As seen from the XRD patterns (Supplementary Fig. 8), NaOH-treated CS showed a slight decrease in the CrI/cellulose values (0.99) compared to untreated CS (1.08), while DEA-treatment decrystallized cellulose more efficiently. The smallest CrI/cellulose value (0.64) was obtained in 40% v/v DEA-treated CS.

In conclusion, the presence of water (best at 60% v/v) is necessary for enhancing the pretreatment efficiency of the DEA system. Enhanced delignification, decreased nonproductive adsorption of cellulase on lignin, and decrystallized cellulose jointly contributed to the high fermentable sugar yield of DEA-treated lignocellulose at a low cellulase loading.

### Single-step N-participated lignin conversion

The processability of the lignin from DEA-based fractionation is another important concern. To our delight, the coproduction of phenolic and N-functionalized monomers was achieved via a tandem strategy coupling amine-based fractionation with hydrogenolysis. Specifically, N-heterocyclic compounds were produced via direct hydrogenolysis of DEA pretreatment liquor, without the need of isolating lignin. Additionally, moderate conditions (ca. 250 °C and 18 bar) and a harmless and green solvent (H₂O) were employed in our work. Until now, considerable research efforts have focused solely on simple dimeric or monomeric lignin models as starting materials to produce N-containing chemicals[26–29]. However, due to the more sophisticated structure of realistic lignin, those tactics that work for model compounds may not hold for real lignin[28,44]. In earlier works (Table 1), lignin was first extracted from lignocellulose using acidic organic solvents and then isolated from the solvents using plenty of water. Subsequently, the isolated lignin was converted to N-containing compounds using two strategies. In the first strategy, lignin was first depolymerized to lignin oil with low molecular weight, and then the resulting oil was converted to benzylamines with amine and Pd/C under argon[28]. Analogously, Ruijten et al. employed an RCF to replace traditional pretreatment for producing refined lignin oil from lignocellulose, followed by sugar isolation and heptane/ethyl acetate extraction to obtain a monomer-enriched fraction of the lignin oil. Finally, the Cu-SiO₂-based amination protocol was applied to convert the fraction into a tertiary amine monomer[45]. The second strategy involves lignin modification prior to depolymerization. For instance, N-modified lignin was produced first by a three-step process (oxidation, oximation and acetylation), and then the modified lignin underwent a photocatalytic reaction to generate nonphenolic arylamine products[46]. Despite the above progress, the current protocols for N-participated lignin conversion still have some shortcomings, such as cumbersome steps, and low N-containing monomer yield.

Herein, the single-step conversion of real lignin to pyridine bases is proposed, whereas in earlier studies, additional refunctionalization was necessary to harvest high-value chemicals from lignin depolymerization products (e.g., low-functionalized aromatics, alkanes, or alcohols)[47–49]. As shown in the insert images in Fig. 2, the lignin-rich liquor from DEA fractionation changed from black to light yellow after hydrogenolysis, while conversely, the NaOH pretreatment liquor became darker (Supplementary Fig. 9). The observation indicated that the lignin-rich stream derived from DEA fractionation was more liable to depolymerization and upgrading than that from NaOH treatment. In addition, there was a decrease in the pH of the DEA pretreatment liquor from the initial value of 11.5 to 9.5 after mild hydrogenolysis, while no noticeable change occurred in the pH of the NaOH pretreatment liquor. This suggested that unreacted amines in the DEA pretreatment stream were consumed and engaged in lignin conversion during the catalytic depolymerization, whereas NaOH base did not take part in the reaction. These findings indicated that depolymerization and amination occurred simultaneously in one pot.

Next, the distribution and yield of the products from hydrogenolysis were analyzed using GC-MS, GC-FID, and GC × GC-MS. After the hydrogenolysis of DEA pretreatment liquor, the mixture was acidified and extracted with ethyl acetate. Monomeric phenolics in the organic phase were identified by GC-MS (Fig. 2a). Among the aromatic monomers, nearly 70% were compounds **A2**, **A4**, **A5**, and **A8**, and the total monophenol yield (15.6 wt%) was over doubled, compared with

**Table 1 | Comparison of N-functional monomer production from realistic lignin in this and previous studies**

| Conversion procedure of lignin into N-containing monomers | Amination reagent | Condition | Yield[a] (wt%) | Ref. |
|---|---|---|---|---|
| I. DEA-based fractionation → II. Hydroprocessing | Diethylamine | II. 250 °C, 4 h, 18 bar H$_2$ | 21.3 (Pyridine bases) | This work |
| I. Acidic organosolv pretreatment → II. Lignin collection from pretreatment liquors → III. Depolymerization into lignin-oil → IV. Amination and further depolymerization | Pyrrolidine | III. 110 °C, 24 h, argon; IV. 120 °C, 20 h, argon | 0.4 (Benzylamines) | 28 |
| I. RCF → II. Sugar, dimers and oligomers removal from lignin oil → III. Amination of lignin-derived monomer in lignin oil | Dimethylamine | I. 235 °C, 3 h, 30 bar H$_2$ III. 210 °C, 16 h, 3 bar H$_2$, 7 bar N$_2$ | ~9[b] (Phenolic tertiary amines) | 45 |
| I. Acidic organosolv pretreatment → II. Lignin recovery from pretreatment liquors → III. Preparation of pre-oxidized lignin → IV. Preparation of acetyl oxime lignin → V. Photocatalytic reaction | Hydroxylamine hydrochloride | V. $\lambda_{max}$ = 365 nm, room temperature, 8 h | 1.9 (Non-phenolic arylamine) | 46 |
| I. Acidic organosolv pretreatment → II. Precipitating lignin from pretreatment liquors → III. Preparation of pre-oxidized lignin → IV. Amination and transformation of lignin | Hydroxylamine hydrochloride | IV. 120 °C, 15 h, N$_2$ | 2.4 (Nitriles and isoxazoles) | 55 |
| I. Preparation of softwood lignin → II. Acidolysis with ethylene glycol → III. Deprotection → IV. Amination → V. Hydrolysis with HBr | NH$_3$ | III. 120 °C, 1 h, 10 bar H$_2$/N$_2$ IV. 160 °C, 6 bar NH$_3$ | 6.4 (Dopamine hydrochloride) | 56 |

[a] N-containing monomer yield based on the weights of lignin.
[b] Depolymerization was initiated with RCF of lignocellulose in this reference, and therefore, the yield here was calculated based on the weight of native lignin.

those from NaOH pretreatment liquor (6.8 wt%). The theoretical maximum yield of monophenols for the lignins isolated from the DEA and NaOH pretreatment liquors was estimated to be 27.5% and 21.0%, respectively, using the nitrobenzene oxidation method. This suggested that the depolymerization efficiency of the lignins in the DEA pretreatment liquors (57%) surpassed that of the ones in the NaOH pretreatment liquors (32%). It is worth noting that carboxylic- or ketone-substituted methoxyphenols or phenols (**A7**–**A11**) were detected, indicating that the C=O groups were not completely reduced and partially survived in hydrogenolysis. There were no phenolic monomers in the DEA pretreatment liquor before hydrogenolysis (Supplementary Fig. 10). Furthermore, lignin-derived phenolic dimers bearing β–O–4′, β–β′, and β–5′ interunit linkages were also present in the hydrogenolysis products of the DEA pretreatment liquor, according to LC-MS analysis (Supplementary Fig. 11).

The introduction of N made another category of products preferentially soluble in acidic environments and therefore remained in the aqueous phase. Pyridine bases were identified in the hydrogenolysis products from the DEA pretreatment liquor, including 5-ethyl-2-methylpyridine (**M5**), 3-ethyl-4-methylpyridine (**M10**), and 2-ethyl-6-isopropylpyridine (**M9**), as shown in the GC-MS profiles in Supplementary Fig. 12, while no such products were detected in those from the NaOH pretreatment liquor and the control experiment (without lignin). However, due to the complexity of the products in the aqueous phase, it was difficult to identify all lignin-derived N-containing products using GC-MS. Alternatively, two-dimensional gas chromatography (GC × GC) was employed, which has higher resolution, larger peak capacity, and higher sensitivity than conventional one-dimensional GC[50]. As a result, more N-containing monomeric products were assigned (Fig. 2b). **M5**, 3-hydroxypyridine (**M6**), and **M10** were the prevailing products, accounting for 70%. Additionally, some phenolic (**B3** and **B6**) and nonphenolic arylamines (**B1**, **B2**, **B4**, and **B5**) were also observed, which likely resulted from the aminated structures of lignin formed in the DEA fractionation. More importantly, the present process achieved an N-containing monomer yield of up to 21.3 wt%, which surpassed the yield reported in the literature (Table 1).

**Structural characterization of the dissolved lignins**

The lignins (DEA-L and AL, respectively) isolated from DEA and NaOH pretreatment liquors were structurally characterized in terms of molecular weight, functional groups, element composition, interunit linkage distribution, and amine incorporation, which could help explain the easy depolymerization of DEA-L in hydrogenolysis. The GPC results showed that both initial DEA-L and AL showed almost identical $M_w$ (2400 Da) (see Supplementary Fig. 13), but the hydrogenolysis products of DEA-L displayed a lower $M_w$ (1000 Da) than those of AL (1500 Da). This indicated that molecular weight was not the main factor that affected lignin reactivity toward catalytic depolymerization.

Increased N content (3.1%) (elemental analysis) and N–H bending vibrations (FTIR analysis) confirmed the N incorporation in DEA-L (Supplementary Fig. 14). 2D HSQC NMR spectra provided more comprehensive structural information of the lignins (Fig. 3), including sidechain and aromatic regions. As shown in the HSQC NMR spectrum of CEL (a typical representative of native lignin) from untreated CS, the substructures A (β–O–4′), B (β–5′), and C (β–β′) were assigned, and β–O–4′ accounted for 61 per 100 aromatic units. The β–O–4′ content declined in both AL and DEA-L, but more β–O–4′ subunits were preserved in DEA-L (37 per 100 aromatic units, A + A′ signals) than in AL (27 per 100 aromatic units), and DEA-L displayed lower β–β′ and β–5′ contents of 5 and 2 per 100 aromatic units, respectively. It is usually agreed that the monomer yield has a positive relation with the β–O–4 ether content in lignin[47,51]. In addition, the $\delta_C/\delta_H$ 40–45/2.6–3.4 ppm signals in the side-chain region, which only appeared in DEA-L, belong to amine and ammonia incorporation. As expected, the amine was

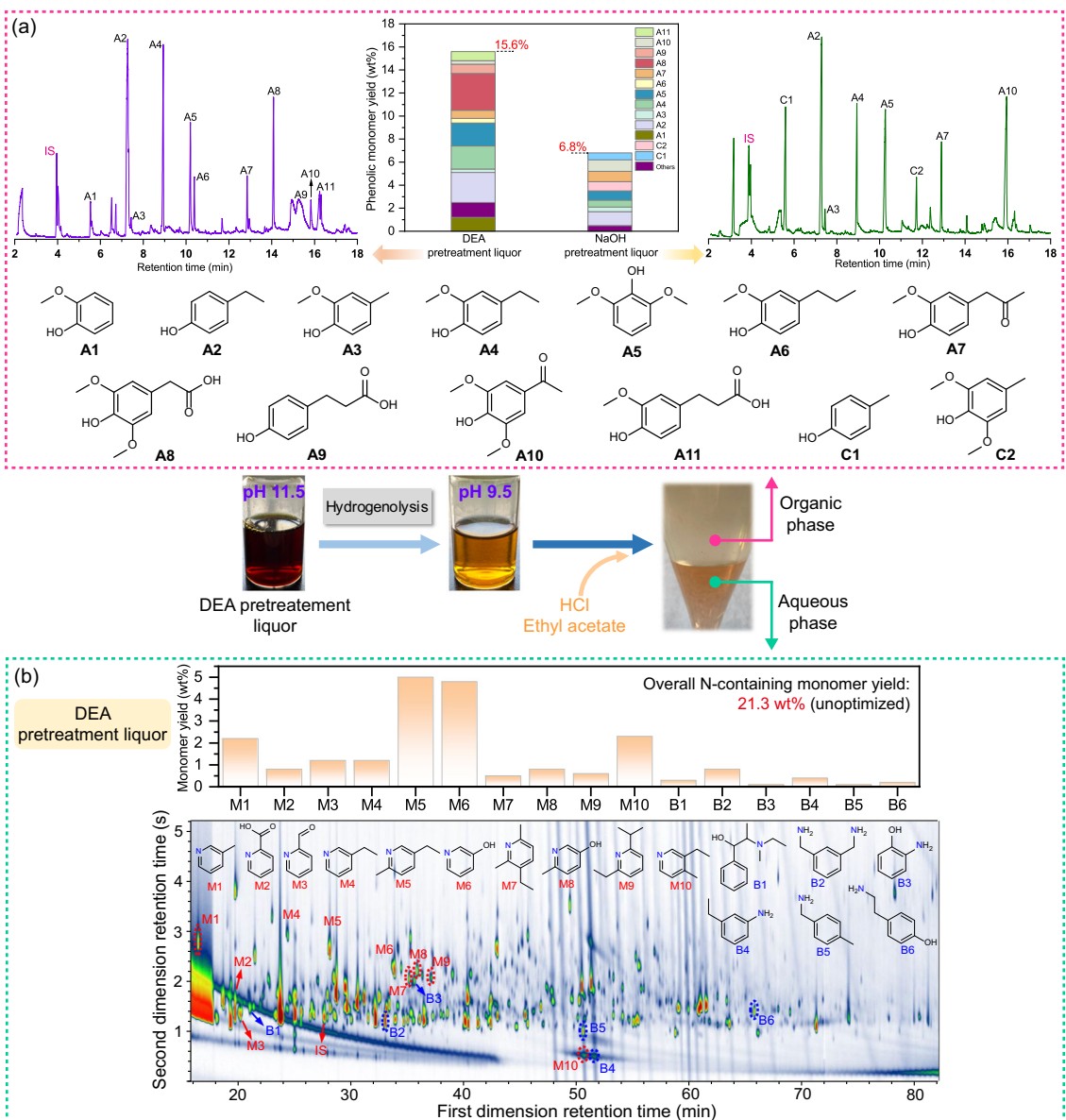

**Fig. 2 | Yield and distribution of monomers in the hydrogenolysis products.**
**a** Phenolic monomers identified by GC-MS in the hydrogenolysis products of the DEA and NaOH pretreatment liquors, and the corresponding product yield quantified by GC-FID. **b** The GC × GC chromatogram of N-containing monomers from the hydrogenolysis of the DEA pretreatment liquor. IS internal standard, DEA diethylamine.

introduced to the α position of β−O−4′ subunits (A) by nucleophilic attack, leading to α-aminated structures (A′). The DFT calculations of the lignin model compound predicted that the bond dissociation energy (BDE) for the β−O−4′ bond was 307 kJ/mol. DEA substitution at the α-position was found to lower the BDE by 10 kJ/mol (Supplementary Note 1). In the aromatic region, DEA-L featured the lowest S unit content (34%) and the highest H unit content (24%) compared to CEL (S: 51%, H: 4%) and AL (S: 42%, H: 10%), suggesting that demethoxylation of S-type lignin probably occurred in DEA fractionation.

Overall, the in situ incorporation of N into lignin was achieved during DEA fractionation. In particular, the blocking of the active α-position formed a stabilized β−O−4′ motif to avoid undesirable repolymerization and lowered the BDE of the β−O−4′ bond. These unique aminated structures combined with a high amount of β−O−4′ subunits and an overall low fraction of C−C interlinkage contributed together to the high activity of DEA-L toward catalytic depolymerization in hydrogenolysis. In addition, aqueous amine is likely to be more favorable for depolymerization of the lignin than NaOH solvents.

## Mechanistic studies

To gain insight into the lignin reaction mechanism during DEA fractionation, a model study was conducted using the most common lignin model compound (guaiacylglycerol-β-guaiacyl ether, GE). Reaction parameters as in 40% DEA fractionation were used for the model study. Afterward, the products were identified by GC-MS (see Supplementary Fig. 15 and Supplementary Note 2). Based on the products from the model compound, an amine-mediated reaction pathway was proposed for the β−O−4′ structure during DEA treatment (Fig. 4). Similar to the reaction of lignin in an alkaline medium[1], the β−O−4′ unit was first transformed into a pivotal intermediate−quinone methide. The intermediate is especially prone to be attacked at α-position by nucleophilic reagents due to its tendency to restore aromaticity. Owing to the strong nucleophilicity of N enhanced by two nearby electron-donating groups, DEA is a strong nucleophile and won the competition with in situ formed lignin nucleophiles, thereby preventing the formation of new C−C interunit linkages. Subsequently, the α-aminated intermediates could experience the scission of β−O−4′ bonds via the

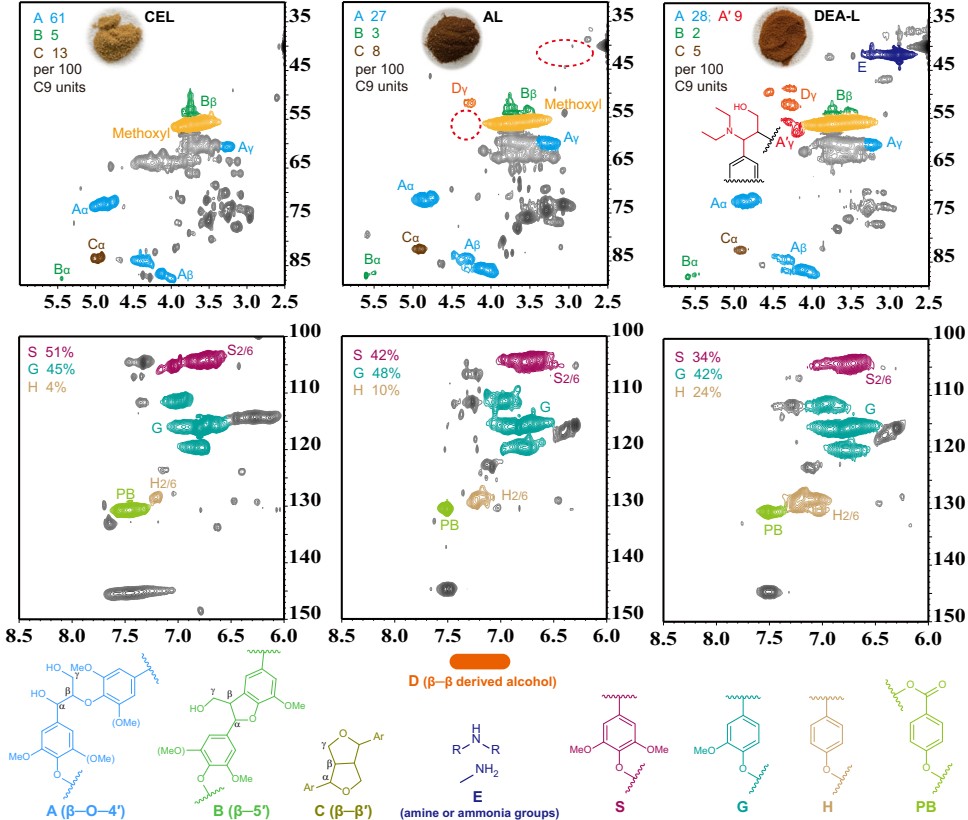

**Fig. 3 | Structural comparison of cellulolytic enzyme lignin (CEL) from untreated corn stover, alkaline lignin (AL) from NaOH pretreatment, and diethylamine-treated lignin (DEA-L) from the fractionation with 40% v/v DEA.** $^1$H-$^{13}$C cross signals and semi-quantitative information on interunit linkages and lignin units.

formation of cyclic nitrogen intermediates. The as-formed intermediates underwent a series of downstream reactions, such as ring-opening, elimination reaction, oxidation, and ammoniation, yielding products such as γ-amino-4-hydroxybenzenepropanol, coniferyl alcohol, 3-(4-hydroxyphenyl)-1-propanol, dihydroferulic acid, and conifer aldehyde. Alternatively, the quinone methide intermediate underwent rearrangement to restore its aromaticity and led to an alkali-stable enol ether motif.

After the fractionation, the stream containing lignin-derived fragments proceeded to the hydrogenolysis stage. The hydrogenolysis products disclosed in Fig. 2 and the oxidized products found in β−O−4′ model studies indicated that the side chains of aromatic substructures were partially oxidized. The products with the side chains retaining (**A7**−**A11**) were generated via the catalytic cleavage of interunit linkages, and on the other side, side-chain cleavage occurred, generating a series of short-chain compounds with C=O groups (Fig. 4) and simple phenolic monomers (**A1**−**A5**). The as-formed compounds bearing C=O groups then condensed with ammonia or primary amines to yield aldimine intermediates. Further condensation between various aldimines formed diamino/diamine imines, which readily underwent cyclization with the elimination of ammonia or amines to produce intermediate tetrahydropyridine[52]. An active hydrogen of the tetrahydropyridine subsequently reacted with an aldimine molecule and then underwent rearrangement by losing ammonia and/or amines to generate the pyridine base products (**M5** and **M10**), which were identified by GC×GC-MS. A plausible reaction pathway with more details between lignin-derived aldehydes and amines/ammonia to produce 5-ethyl-2-methylpyridine is shown in Supplementary Fig. 16. Notably, DEA could be oxidized and then undergo cope reactions to give substituted hydroxylamine and alkene. Ammonia or primary amines were formed via the decomposition of substituted hydroxylamine at high temperatures.

To further confirm the hypothesis that aromatic compounds with oxidized end groups can be transformed to pyridine bases under aqueous DEA and hydrogenolysis conditions, four lignin model compounds bearing C=O groups, including 4-hydroxyacetophenone, 4-hydroxy-3-methoxyphenylpyruvic acid, 4-hydroxyphenylacetic acid, and 4-hydroxy-3-methoxycinnamaldehyde, were investigated in the similar reaction environment. From Supplementary Note 3, products such as phenol, guaiacol, *p*-cresol, 4-ethylphenol, 2-methoxy-4-methylphenol, and 4-ethylguaiacol were detected from the side-chain cleavage of the model compounds. More importantly, numerous pyridine base products were generated. The results revealed that pyridine bases were synthesized via the pathway of the scission of the oxidized side chain followed by the reactions between the cleaved side chain and amines.

Finally, the role of Cu(OAc)$_2$ was unmasked through a comparison of the products obtained in hydrogenolysis with and without Cu(OAc)$_2$. As shown in Supplementary Fig. 17, the hydrogenolysis liquor without Cu(OAc)$_2$ appeared darker in color and had a higher pH (10.7), compared to those with Cu(OAc)$_2$, suggesting that a copper catalyst facilitated lignin depolymerization and the amination of the products. GC-MS analysis revealed that the hydrogenolysis liquor without Cu(OAc)$_2$ contained fewer **A2**, **A4**, and **A5** compounds but more compounds bearing C=O groups (**A7** and **A9**). In addition, Cu(OAc)$_2$ significantly increased the content of pyridine bases in the hydrogenolysis liquor. From the synthesis mechanism of substituted pyridines above, we speculated that Cu(OAc)$_2$ played an important role in cleaving the oxidized side chains of the lignin. The cleaved side chains were subsequently aminated to yield substituted pyridines. In the presence of Cu(OAc)$_2$, a similar cleavage of the C−C bond was observed in the reaction of different lignin models with amines/ammonia[53]. Additionally, a Cu-based catalyst has been used to catalyze the amination of monomers and dimers in lignin oil[45]. On the other

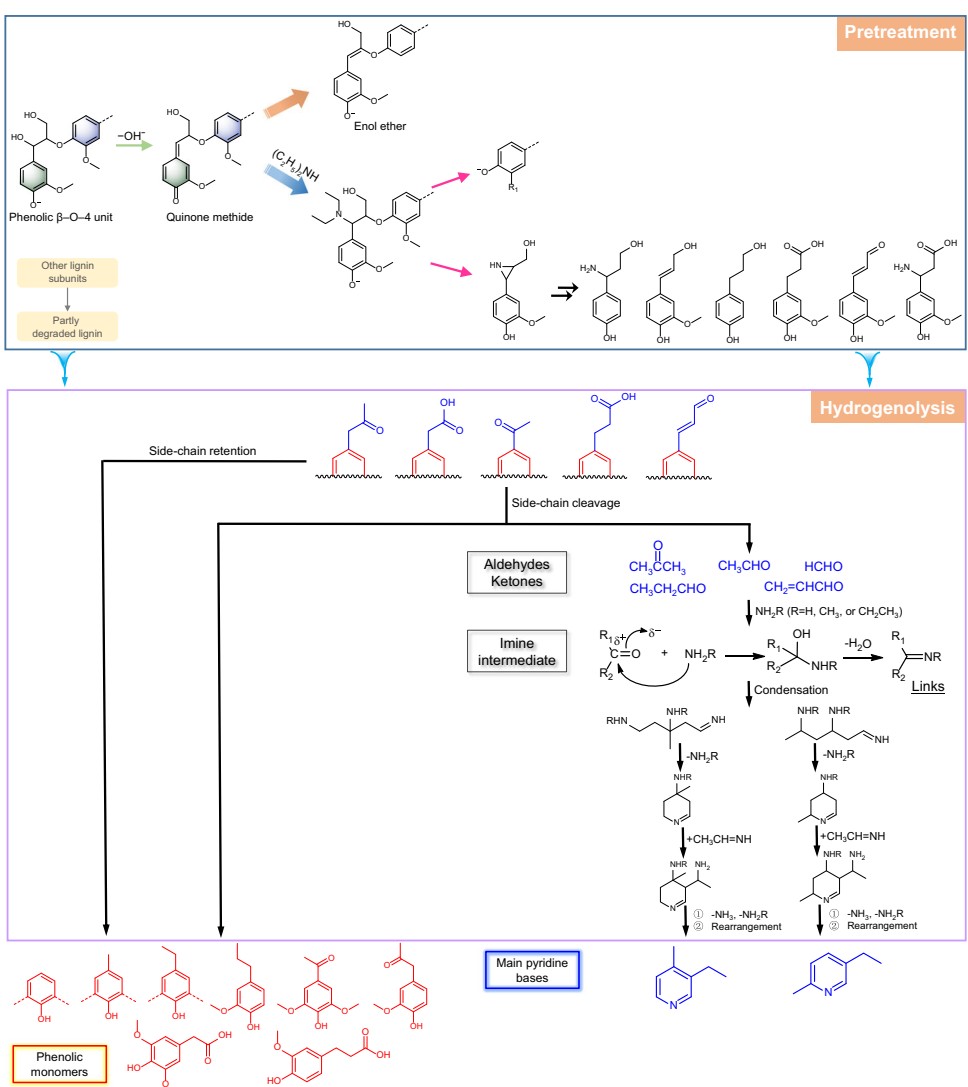

**Fig. 4 | Mechanistic studies.** Reaction mechanism in diethylamine-based pretreatment for β−O−4′ motif and plausible synthesis pathway for pyridine bases from lignin.

hand, Cu(OAc)$_2$ may be involved in the activation of substrates that lead to the synthesis of pyridine derivatives.

## Discussion

In the carbohydrate-first strategies, the liberation of lignin from the lignocellulose matrix is the primary target to accelerate carbohydrate valorization. However, harsh processing conditions were generally applied to reach extensive delignification, which in turn negatively affected the reactivity of the resultant lignin due to condensed structures and thereby the yield of lignin-to-aromatic conversion. To avoid lignin repolymerization, moderate conditions of 130 °C and 1 h were employed for DEA treatment in our study. Under such mild conditions, lignin was isolated in high yield, because aqueous DEA facilitated lignin migration and dissolution, and lignocellulose ultrastructure deconstruction. More importantly, the incorporation of N into lignin substructures, especially at the α-active position of the β−O−4′ motif, efficiently suppressed lignin condensation and preserved more β−O−4′ structures. Therefore, high-reactivity lignin with easily degradable structures enabled an efficient catalytic depolymerization in hydrogenolysis to produce value-added monomers in high yield.

The lignin-first concept pays much more attention to the optimal utilization of lignin. The addition of extra protective reagents during biomass fractionation, e.g., aldehydes, to trap the benzyl carbocation intermediate is the state-of-the-art stabilization approach[11,18]. Although

lignin with protection groups prominently promoted lignin-derived monomer production, the lignocellulosic substrates generated in lignin-first biorefineries cannot be enzymatically hydrolyzed as effectively as those from carbohydrate-first strategies. Our fractionation strategy not only facilitated catalytic lignin conversion but also improved the enzymatic hydrolyzability of carbohydrates due to high delignification, decrystallization of cellulose, and sharply reduced nonproductive adsorption of cellulase on the in situ N-modified lignin. Furthermore, the lignin-rich DEA pretreatment liquor could enhance the enzymatic hydrolysis performance of lignocellulosic substrates.

To expand the product portfolio and improve the economic viability of biomass refineries, the production of lignin-derived N-containing chemicals has become a rising research interest. Pyridine bases are widely used heterocycles in the fields of dyes, polymers, and medicinal chemistry[54]. In this study, pyridine bases were synthesized in high yield from real lignin. Compared with other complicated N-participated lignin conversion pathways, pyridine bases were produced via a simple tandem strategy in this study, DEA-based fractionation of lignocellulose followed by direct hydrogenolysis of the lignin-rich stream. Furthermore, a satisfactory yield (21.3 wt%) of N-containing monomeric compounds based on realistic lignin was obtained, which was significantly higher than the yield of <10 wt% in previous studies[28,45,46,55,56]. The pyridine bases were formed in two steps. First, the reactions between the aldehydes and ketones derived

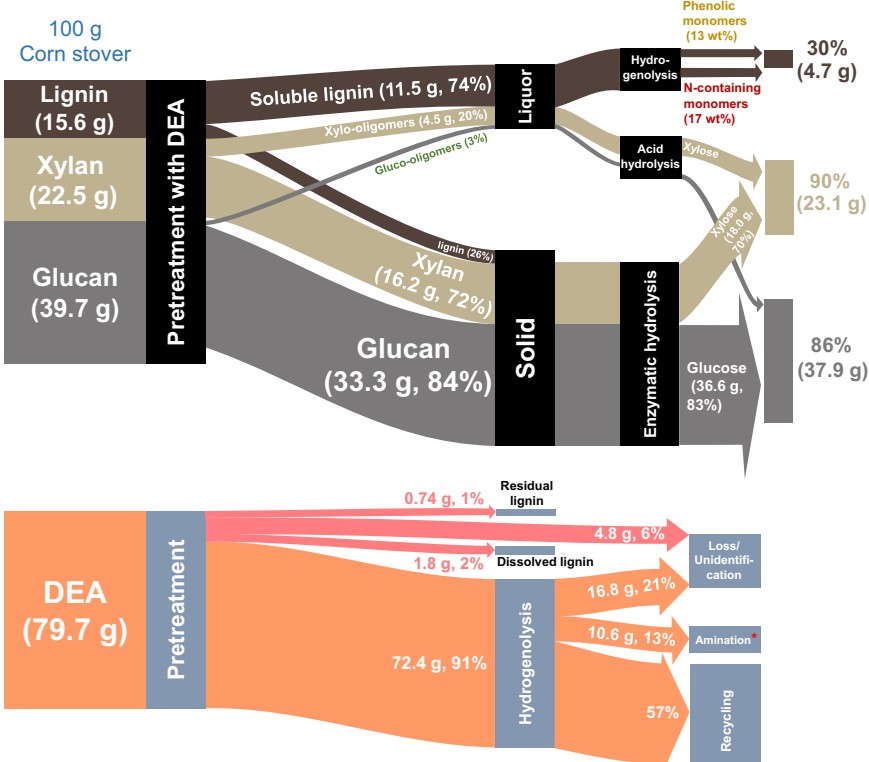

**Fig. 5 | Mass balance flow of polysaccharide (glucan and xylan), lignin fractions, and DEA on a basis of 100 g of biomass.** Pretreatment condition: 40% v/v DEA, 30 wt% biomass loading, 130 °C, and 1 h. CTec2 loading used here is 15 FPU/g glucan. *All aminated compounds in the hydrogenolysis liquors. DEA diethylamine.

from the lignin side chains and ammonia/amines generated imine intermediates. These intermediates then undergo condensation, cyclization, and rearrangement to yield the target products. The majority (70%) of the final products were identified as 2-methyl-5-ethylpyridine, 3-ethyl-4-methylpyridine, and 3-hydroxypyridine.

The DEA process in this study achieved the efficient covalorization of lignin and polysaccharide fractions (see mass balance flow in Fig. 5), which reconciled the dilemma of carbohydrate- and lignin-first biorefineries. The fractionation technology with a DEA-water yielded 30 wt% lignin monomers, 90% xylose, and 86% glucose. Approximately 13% glucan and 8% xylan were not recovered during the DEA pretreatment. The common degradation products of carbohydrates reported in other works[57,58], such as furfural, 5-hydroxymethylfurfural (5-HMF), and levulinic acid, were not detected in the DEA pretreatment liquor (Supplementary Fig. 18). To investigate potential side reactions between xylan and amines, we treated xylose in a 40% v/v DEA solution (Supplementary Note 4). The GC-MS analysis confirmed the presence of cyclopentene derivatives, furan derivatives, and N-heterocyclic compounds in the product mixture. Apart from producing small molecule compounds, brown nitrogenous polymers may be generated through Maillard reactions between carbohydrates and amines[59].

In addition to the mass flow of lignocellulosic components, a Sankey chart of DEA was established, as presented in Fig. 5. During the processing of CS, ~6% of DEA was lost due to its low boiling point. Another 1% and 2% of DEA were consumed in the amination of residual lignin and dissolved lignin, respectively. Approximately 91% of the unreacted DEA in the pretreatment liquor proceeded to the next step. In hydrogenolysis, 13% of DEA participated in the amination reactions, yielding a large amount of N-functionalized compounds. The remaining DEA could be recycled from the phenol and pyridine products via rectification, as confirmed by the Aspen simulation (Supplementary Fig. 19).

We next conducted a preliminary techno-economic analysis (TEA) of the N-participated lignin valorization to evaluate the economic viability of the proposed method, with a particular focus on determining the minimum pyridine selling price (MPSP)[3]. Based on our experimental design, an Aspen process flow diagram for pyridine base production from lignin is presented in Supplementary Fig. 19. This model integrates two main steps: (1) the catalytic depolymerization and amination of lignin; (2) the separation and purification of the products. The MPSP in the simulation system was $2.8/kg (Supplementary Table 4), lower than the market prices of $3.0–3.5/kg[60]. The result indicates the significant potential of our innovative lignin valorization approach to drive the development of cost-effective biorefineries. However, it should be noted that the TEA solely represents an initial assessment since the proposed technology is still in its infancy.

Finally, we also noticed that pure ethylenediamine (EDA) was used in some studies to achieve high enzymatic digestibility of CS and produce lignin with extraordinary aqueous solubility[61-63]. However, further processing opportunities of the lignin produced from the pretreatments were not studied. Moreover, DEA is nearly twice as inexpensive as EDA, and an aqueous solution of DEA was used for the fractionation in this study, which further reduced the cost of the solvent.

## Methods

### High-solid pretreatment of corn stover

Corn stover pretreatment was carried out in a 100-ml stainless steel reactor with a Teflon lining. First, 5.4 g CS was mixed with a pretreatment reagent at a biomass loading of 30 wt%. The pretreatment reagents used here included pure DEA, aqueous solution of DEA at different concentrations, deep eutectic solvent (DES) composed of choline chloride and glycerol (1:2, molar ratio)[64], and ammonia solution (≥28%). The reactor was heated to 130 °C and kept at the

temperature for 1 h in a drying oven. After completing the reaction, the reactor was cooled down, and the resultant mixture was rinsed out with water (50 ml × 3). The solid and the liquid were separated using a filter bag, and the solid was dried at 60 °C to a moisture content below 10%. The lignin-rich liquor was collected and named as pretreatment liquor.

## Enzymatic hydrolysis of the pretreated corn stover substrates
Enzymatic hydrolysis of the pretreated substrates was performed at 2% w/v glucan loading with a total volume of 20 ml at 50 °C, pH 4.8 (50 mM NaAc-HAc buffer), and 150 rpm. Two mg of tetracycline was added to avoid bacterial contamination. Commercial enzymes CTec2 were added at the required dosages in 8 FPU/g glucan. Following the enzymatic hydrolysis for 72 h, an aliquot (500 µl) of the hydrolysate was taken, centrifuged, and filtered through a 0.22-µm membrane filter. Finally, the glucose and xylose concentrations in the supernatant were analyzed using HPLC with an HPX-87H column.

## Typical procedure for the hydrogenolysis of pretreatment liquors
The hydrogenolysis of pretreatment liquors was performed under an $H_2$ pressure of 18 bar. Specifically, pretreatment liquor (20 ml), 10% Pd/C (50 mg), and $Cu(OAc)_2$ (30 mM) were placed in a high-pressure reactor (50 ml) and stirred at 400 rpm at 250 °C for 240 min. After that, the reactor was rapidly cooled down in an ice-water bath. The mixture from the hydrogenolysis was acidified to pH = 2 with 5% hydrochloric acid, and ethyl acetate was added to extract phenolic monomers. The organic phase was dried with a rotary evaporator, and then 3 ml of acetone and 0.5 ml of n-decane (1.0 mg/ml in methanol, internal standard) were added to completely dissolve the dry residue. Lignin-derived phenolics were identified by GC-MS and quantified by GC-FID. For the aqueous phase after extraction, water was removed first with a rotary evaporator, and the products were dried in an oven at 55 °C and then dissolved in 1 ml of acetone and 0.3 ml of internal standard. Before GC × GC-TOF-MS analysis, silylation was conducted to increase product volatility. In brief, 150 µl of the sample solution was mixed with 500 µl N-methyl-N-trimethyl-silyl-trifluoroacetamide (MSTFA) and 500 µl pyridine, and the mixture was held at room temperature for at least 2 h prior to analysis.

## Data availability
The experiment data generated in this study are provided in the Source Data file. Any other relevant data are available from the authors upon request. Source data are provided with this paper.

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

## Acknowledgements

This study was financially supported by the National Key Research and Development Program of China Plan (grant no. 2021YFC2102700), the National Natural Science Foundation of China (grant nos. 21978106 and U21A20309), and the Fundamental Research Funds for the Central Universities (grant no. 2022ZYGXZR108). H.L. received the fund-ing above.

## Author contributions

L.X. conceived the idea, designed and performed the experiment. M.C. participated in the GC-MS and GC-FID characterization. J.Z. performed the techno-economic analysis. Z.L. contributed the nitrobenzene oxi-dation of lignins. L.X. and H.L. co-wrote the manuscript. Y.P., D.Y., H.L., X.P., and X.Q. helped in writing—reviewing and editing and funding acquisition. H.L. and X.Q. supervised the project. S.-Y.L. and X.P. helped with the data analyses and experiment design. All authors reviewed and approved the final manuscript.

## Competing interests

The authors declare no competing interests.
