## [Peer Review File · Nature Communications]

Reviewers' Comments:

Reviewer #1:

Remarks to the Author:

In this work, a new approach using aqueous diethylamine for high-solid fractionation and subsequent tandem amine-based fractionation/mild hydroprocessing strategy for the conversion of lignin into pyridine bases is communicated. The methodology successfully addresses the trade-off between carbohydrate utilization efficiency and lignin-to-chemical conversion yield, opening up new avenues for lignin utilization. In my opinion, the work is potentially transformative for the field of lignocellulosic biorefineries and I am pleased to recommend its acceptance after addressing the following minor issues.

Many organonitrogen chemicals are produced from lignin. These chemicals have different structures. It would be good for the authors to provide possible reaction pathways that lead to the production of these compounds (at least some major products).

Although DEA is not a very expensive organic base, it is still costly if the utilization efficiency is low. Would DEA be able to be recycled in a second batch use?

As the study aims to contribute to sustainable and cost-effective biorefineries, it would be beneficial to include a discussion on the economic viability of the proposed method and its potential environmental impact.

On a related note, a simple/brief life-cycle assessment or techno-economic analysis would provide valuable information on the process's sustainability and commercial feasibility.

Finally, related review articles on making organonitrogen chemicals from biomass may be included in the reference (i.e., Expanding the boundary of biorefinery: organonitrogen chemicals from biomass, *Accounts of Chemical Research*, 2021, 54, 1711-1722)

Reviewer #2:

Remarks to the Author:

This manuscript entitled "Aqueous amine enables sustainable monosaccharide, monophenol and pyridine base coproduction in lignocellulosic biorefineries" exhibited an interesting work in effectively valorizing the entire lignocellulosic biomass. More importantly, this work provided the first workable deal for converting real lignin into pyridine bases, which could propel the sustainable production of high value-added chemicals. It could be improved to be in the top 5% with appropriate revisions. My comments are listed below.

1. Fig. 1e. Two commercial enzyme preparations, celluclast 1.5L and Ctec 2, were used in the enzymatic hydrolysis of the pretreated biomass. Selecting same cellulase in all saccharification process is the better choice.

2. Fig. 1f and Supplementary Fig. 9. The authors highlighted that residual lignin in the biomass treated with DEA fractionation showed lower binding ability with cellulase compared to those treated with traditional pretreatment. However, the cellulase adsorption was compared only in the DEA-RL and CEL, without the lignin sample from traditional pretreatment. Please clarify that.

3. Fig. 2. To make the protocol easy to understand, an image for organic and aqueous phase after extraction should be provided.

4. Lines 346-349. Solid-state lignin depolymerization is interesting. However, no experimental evidence was provided in this manuscript or the cited literature to support its contribution for uncondensed structures. Please clarify that.

5. Lines 383-390. The model experiment (Lines 393-403) offered evidence for the synthesis of pyridine bases between the cleaved lignin side chains and amines. However, the detailed pathway presented in Fig. 4 was mainly based on the published mechanism study of pyridine products, which was not proved in this work. Therefore, the related expression (lines 28-29) should be deleted.

6. What is the role of Cu(OAc)₂ in the hydroprocessing?

Reviewer #3:

The manuscript entitled "Aqueous amine enables sustainable monosaccharide, monophenol and pyridine base coproduction in lignocellulosic biorefineries" by Qiu and coworkers describe a fractionation approach in which secondary aliphatic amine is used as reagent and base to produce monophenols and substituted pyridines.

The approach is rather novel and the authors should have credit for valorizing all parts of the lignocellulose. However, there are some weaknesses that need to be addressed by the authors.

First of all, lignin-first is not defined as lignin has a higher value than the other parts of the wood. The authors should look into the guidelines paper written a couple of years back to find an appropriate definition (this paper should be cited):

"lignin-first" biorefining approach, in which lignin valorisation is considered in the design phase.

This has been demonstrated the last couple of years where lignin has been valorized and pulp has been upgraded to for instance dissolving pulp. Thus, it is not correct that carbohydrates generally are sacrificed.

When it comes to the results, it is important to understand what the 13 wt% yield monophenols correspond to. The authors should determine the beta-ether content of the feedstock.

When it comes to the N-containing aromatic compounds, the authors should determine the mass balance stemming from DEA and the biomass. How much DEA is incorporated into valuable products and how much can be recovered and re-used? DEA is missing in Fig 5.

The authors propose that the amine is substituting the benzylic alcohol, which sounds reasonable; it is not really understandable why the authors perform the hydrogenolysis not on the mixture but on certain models? What happens if product mixture (disclosed in Fig S18) would be used?

Furthermore, even hemicellulose could react with the amine, and this should be studied (furfural, 5-HMF or levulinic acid).

This reviewer does not agree that hydrogenolysis performed in water would have any advantages over organic solvents. Rather the contrary from an industrial point of view where HDO reactions are performed using carrier liquids to control reactivity.

Manuscript ID: NCOMMS-23-32549

Manuscript Title: Aqueous amine enables sustainable monosaccharide, monophenol, and pyridine base coproduction in lignocellulosic biorefineries

Response to Reviewer #1:

In this work, a new approach using aqueous diethylamine for high-solid fractionation and subsequent tandem amine-based fractionation/mild hydroprocessing strategy for the conversion of lignin into pyridine bases is communicated. The methodology successfully addresses the trade-off between carbohydrate utilization efficiency and lignin-to-chemical conversion yield, opening up new avenues for lignin utilization. In my opinion, the work is potentially transformative for the field of lignocellulosic biorefineries and I am pleased to recommend its acceptance after addressing the following minor issues.

1. Many organonitrogen chemicals are produced from lignin. These chemicals have different structures. It would be good for the authors to provide possible reaction pathways that lead to the production of these compounds (at least some major products).

Response: We appreciate your valuable suggestions. We have provided the reaction pathways for major organonitrogen products, including 5-ethyl-2-methylpyridine, 3-ethyl-4-methylpyridine, and 3-hydroxypyridine, in Fig. 4. These pathways involve condensation, cyclization, elimination, and rearrangement reactions between the oxidized side chain in lignin and amine/ammonia. In response to the comment, the reaction pathway with more details between lignin-derived aldehydes and amine/ammonia to produce 5-ethyl-2-methylpyridine was added as Supplementary Fig. 16.

Modification:

“A plausible reaction pathway with more details between lignin-derived aldehydes and amine/ammonia to produce 5-ethyl-2-methylpyridine is shown in Supplementary Fig. 16.” was added in Lines 362–364 in the revised manuscript.

Supplementary Figure 16. Possible pathway to produce 5-ethyl-2-methylpyridine from the reaction between lignin-derived aldehydes and amine/ammonia. I: 5-Ethyl-2-methylpyridine; II: Aldimine; III: Amino imine/Amine imine; IV: Diamino imine/Diamine imine; V: Tetrahydropyridine.

2. Although DEA is not a very expensive organic base, it is still costly if the utilization efficiency is low. Would DEA be able to be recycled in a second batch use?

Response: We fully agree with the comment that recycling DEA is important to enhance the commercial feasibility of the process. Certainly, DEA can be recycled via rectification due to the difference in volatility of DEA with other chemicals [In, H. et al. *METHOD FOR RECOVERING AMINE FROM AMINE-CONTAINING WASTE WATER*. Patent US 8,545,704 B2 (2013)]. As shown in the Aspen flowsheet (Supplementary Fig. 19), the outlet material from the reactor is directed into the primary separation tower (B6). The upper material from the primary separation tower contains a substantial amount of DEA and pyridine. It is then introduced into the DEA recovery tower (B8), where DEA is separated and recovered from the tower's top.

Modification:

“In addition to the mass flow of lignocellulosic components, a mass balance of DEA was established, as presented in Fig. 5b. During the processing of CS, approximately 6% of DEA was lost

due to its low boiling point. Another 1% and 2% of DEA were consumed in the amination of residual lignin and dissolved lignin, respectively. Approximately 91% of the unreacted DEA in the pretreatment liquor proceeded to the next step. In hydrogenolysis, 10% of DEA participated in the amination reactions, yielding N-functionalized monomers and oligomers. The remaining DEA could be recycled from the phenol and pyridine products via rectification, as confirmed by the Aspen simulation (Supplementary Figure 19).” was added in Lines 446–453 in the revised manuscript.

Supplementary Figure 19. Aspen Plus process flow diagram of the N-participated lignin valorization to pyridine bases and phenols. The separation and recycling were as follows: The outlet material from the reactor is directed into the primary separation tower (B6). From the primary separation tower kettle, phenolic wastewater is discharged. Following a 5-stage extraction process using methyl isobutyl ketone solvent (B3), the phenolic solvent is directed to the solvent recovery tower for the extraction of crude phenol. The upper material from the primary separation tower contains a substantial amount of DEA and pyridine. It is then introduced into the DEA recovery tower (B8), where DEA is separated and recovered from the tower's top. The kettle discharge from the tower contains water and pyridine, which is subsequently fed into the azeotropic distillation unit, using *n*-propyl formate as the azeotrope agent. This process ultimately accomplishes the separation of water from pyridine.

3. As the study aims to contribute to sustainable and cost-effective biorefineries, it would be beneficial to include a discussion on the economic viability of the proposed method and its potential environmental impact.

On a related note, a simple/brief life-cycle assessment or techno-economic analysis would provide valuable information on the process's sustainability and commercial feasibility.

Response: We appreciate your valuable comment, and agree that a techno-economic analysis (TEA) could help us to assess the process's commercial feasibility. As suggested, we conducted a preliminary TEA for the process. It turned out that the minimum selling price for pyridine bases in the proposed technology was \$2.8/kg, which was lower than the market price (\$3.03.5/kg).

Modification:

“We next conducted a preliminary techno-economic analysis (TEA) of the N-participated lignin valorization to evaluate the economic viability of the proposed method, with a particular focus on determining the minimum pyridine selling price (MPSP).³ Based on our experimental design, an Aspen process flow diagram for pyridine base production from lignin is presented in Supplementary Fig. 19. This model integrates two main steps: (i) the catalytic depolymerization and amination of lignin; (ii) the separation and purification of the products. The MPSP in the simulation system was \$2.8/kg (Supplementary Table 4), lower than the market prices of \$3.0–3.5/kg.⁶² The result indicates the significant potential of our innovative lignin valorization approach to drive the development of cost-effective biorefineries. However, it should be noted that the TEA solely represents an initial assessment since the proposed technology is still in its early stages.” was added in Lines 458–468 in the revised manuscript.

Supplementary Table 3. Mass flow of key streams.

Component	Units	S1	S3	S4	S13	S7	S14	S15	PY
Lignin	kg/hr	700	-	454.983	-	-	-	-	-
DEA	kg/hr	-	784.819	685.436	-	685.362	675.98	31.962	-
Pyridine	kg/hr	-	-	139.995	0.00047	139.505	0.05946	139.566	139.488
Phenol	kg/hr	-	-	104.996	102.005	2.94737	-	2.97247	1.14177
H ₂ O	kg/hr	-	18835.6	18914.2	-	377.299	6.91979	470.898	-

Supplementary Table 4. Techno-economic analysis (TEA) of the lignin valorization to pyridine bases and phenols.

	Value
Annual production/MMkg	1.94

Total capital cost/MMS	31.65
Total operation cost/MMS/yr	2.1
Raw material/MMS\$/yr	1.5
Utilities/MMS\$/yr	0.54
Rate of return/%	10
Minimum pyridine selling price/\$/kg*	2.8

*MPSP was calculated based on the coproduction of phenols with a selling price of \$1.3/kg.

4. Finally, related review articles on making organonitrogen chemicals from biomass may be included in the reference (i.e., Expanding the boundary of biorefinery: organonitrogen chemicals from biomass, Accounts of Chemical Research, 2021, 54, 1711-1722)

Response: Thanks to the reviewer's recommendation. This reference is useful and has been cited in the Introduction.

Modification:

“N-participated lignin conversion, which targets sustainable heteroatom-functionalized monomer production, is of great importance to expanding the product pool to meet value-added biorefinery demand.” was changed to “N-participated lignin conversion, which targets sustainable heteroatom-functionalized monomer production, is of great importance to expanding the product pool of lignin to meet value-added biorefining demands.²⁶” in Lines 77–79 in the revised manuscript.

Response to Reviewer #2:

This manuscript entitled “Aqueous amine enables sustainable monosaccharide, monophenol and pyridine base coproduction in lignocellulosic biorefineries” exhibited an interesting work in effectively valorizing the entire lignocellulosic biomass. More importantly, this work provided the first workable deal for converting real lignin into pyridine bases, which could propel the sustainable production of high value-added chemicals. It could be improved to be in the top 5% with appropriate revisions. My comments are listed below.

1. Fig. 1e. Two commercial enzyme preparations, celluclast 1.5L and Ctec 2, were used in the enzymatic hydrolysis of the pretreated biomass. Selecting same cellulase in all saccharification process is the better choice.

Response: Thanks to the valuable suggestion. we agree that using the same cellulase in all

saccharification processes would make the results more comparable. Therefore, we re-ran the enzymatic hydrolysis of different substrates using the same cellulase (CTec2) at a cellulase dosage of 8 FPU/g glucan. Fig. 1e has been updated in the revised manuscript.

Modification:

Fig. 1e. Glucose and xylose yield of untreated CS and CS treated with different pretreatment reagents. The enzyme dosage was 8 FPU CTec2/g glucan.

2. Fig. 1f and Supplementary Fig. 9. The authors highlighted that residual lignin in the biomass treated with DEA fractionation showed lower binding ability with cellulase compared to those treated with traditional pretreatment. However, the cellulase adsorption was compared only in the DEA-RL and CEL, without the lignin sample from traditional pretreatment. Please clarify that.

Response: We appreciate the professional comment, and we did not provide the adsorption behaviors of cellulase on the residual lignin from traditional pretreatment in the previous manuscript. In our recently published article [Xu, L. et al. *Unveiling the role of long-range and short-range forces in the non-productive adsorption between lignin and cellulases at different temperatures. J. Colloid Interf. Sci.* 647, 318–330 (2023)], we determined the adsorption capacity of cellulase on lignin extracted from organosolv, hydrothermal, and alkali-pretreated corn stover to be 224.3, 153.9, and 162.6 ng/cm², respectively, using QCM-D. We have included a comparison of these data in the revised manuscript.

Modification:

“The maximum enzyme adsorption on the lignins isolated from the CS treated with acidic ethanol/water solution, hot water, and NaOH was 224.3, 153.9, and 162.6 ng/cm², respectively, in our previous work,²⁴ which are pronouncedly larger than that on DEA-RL.” was added in Lines 177–180 in the revised manuscript.

“Based on these findings, the adsorption dissimilarity of cellulase on lignin from DEA and traditional pretreatment was proposed (Supplementary Fig. 9).” was changed to “Based on these findings, we proposed the dissimilarity in adsorption of cellulase on lignins obtained from DEA and traditional pretreatment methods (Supplementary Fig. 7). The increased hydrophilicity of DEA-RL significantly reduces its binding ability with cellulase compared to lignins from other pretreatment methods.” in Lines 180–183 in the revised manuscript.

3. Fig. 2. To make the protocol easy to understand, an image for organic and aqueous phase after extraction should be provided.

Response: We appreciate the valuable suggestion. We fully agree that a photograph is more visually informative, and we have added the corresponding images in Fig. 2.

Modification:

Fig. 2 Yield and distribution of monomers in the hydrogenolysis products. (a) Phenolic monomers identified by GC-MS in the hydrogenolysis products of the DEA and NaOH pretreatment liquors, and the corresponding product yield quantified by GC-FID. **(b)** The GC \times GC chromatogram of N-containing monomers from the hydrogenolysis of the DEA pretreatment liquor.

4. Lines 346-349. Solid-state lignin depolymerization is interesting. However, no experimental evidence was provided in this manuscript or the cited literature to support its contribution for uncondensed structures. Please clarify that.

Response: We deeply regret the lack of rigor in the previous version on the solid-state lignin depolymerization. After careful reading and discussion of the reference [Li, N. *et al.* *An uncondensed*

lignin depolymerized in the solid state and isolated from lignocellulosic biomass: a mechanistic study. Green Chem., 2018, 20, 4224], we acknowledged that DEA behaves differently from the LiBr system reported in the reference. DEA depolymerizes and dissolves lignin to about 74%, whereas lignin is not dissolved in the LiBr system. Therefore, the lignin depolymerization in our work is different from that in the reference, and we have removed the related content from the revised manuscript.

Modification:

“In a ‘dry-to-dry’ process, solid-state depolymerization of lignin (similar to the situation in the reported lithium bromide trihydrate system) also contributed to the formation of uncondensed lignin structures.⁴⁷” was deleted in the revised manuscript.

“More importantly, the incorporation of N into lignin substructures, especially at the α -active position of the β -O-4' motif, as well as the restriction of the solid-state pretreatment system to the mobility of reactive lignin intermediates,^{47, 49} efficiently suppressed lignin condensation and preserved more β -O-4' structures.” was changed to “More importantly, the incorporation of N into lignin substructures, especially at the α -active position of the β -O-4' motif, efficiently suppressed lignin condensation and preserved more β -O-4' structures.” in Lines 401–403 in the revised manuscript.

5. Lines 383-390. The model experiment (Lines 393-403) offered evidence for the synthesis of pyridine bases between the cleaved lignin side chains and amines. However, the detailed pathway presented in Fig. 4 was mainly based on the published mechanism study of pyridine products, which was not proved in this work. Therefore, the related expression (lines 28-29) should be deleted.

Response: Thanks for your careful review. We have deleted the related expression in the Abstract.

Modification:

“The process is the first example of transforming realistic lignin into pyridine bases in high yields, involving condensation, cyclization, elimination and rearrangement reactions between the oxidized side chain in lignin and amine/ammonia, which opens up a new way for the efficient upgrading of lignocellulose.” was changed to “The process is the first example of transforming real lignin into pyridine bases in high yield, which opens a new approach to the efficient valorization of lignocellulose.” in Lines 27–28 in the revised manuscript.

6. What is the role of Cu(OAc)₂ in the hydroprocessing?

Response: We greatly appreciate this insightful question. In response to this inquiry, we

conducted hydrogenolysis experiments with and without Cu(OAc)₂, allowing us to make a comparative assessment of the resulting products. In conclusion, the presence of Cu(OAc)₂ promotes the cleavage of oxidized lignin side chains and the production of substituted pyridines.

Modification:

“Finally, the role of Cu(OAc)₂ was unmasked through a comparison of the products obtained in hydrogenolysis with and without Cu(OAc)₂. As shown in Supplementary Fig. 17, the hydrogenolysis liquor without Cu(OAc)₂ appeared darker in color and had a higher pH (10.7), compared to those with Cu(OAc)₂, suggesting that a copper catalyst facilitated lignin depolymerization and the amination of the products. GC-MS analysis revealed that the hydrogenolysis liquor without Cu(OAc)₂ contained fewer **A2**, **A4**, and **A5** compounds but more compounds bearing C=O groups (**A7** and **A9**). In addition, Cu(OAc)₂ significantly increased the content of pyridine bases in the hydrogenolysis liquor. From the synthesis mechanism of substituted pyridines above, we speculated that Cu(OAc)₂ played an important role in cleaving the oxidized side chains of the lignin. The cleaved side chains were subsequently aminated to yield substituted pyridines. In the presence of Cu(OAc)₂, a similar cleavage of the C–C bond was observed in the reaction of different lignin models with amines/ammonia.⁵⁷ Additionally, a Cu-based catalyst has been used to catalyze the amination of monomers and dimers in lignin oil.⁴⁹” was added in Lines 380–392 in the revised manuscript.

Supplementary Figure 17. Comparison of the hydrogenolysis products with and without Cu(OAc)₂. The structures of phenolic monomer (A1–A11) and substituted pyridine (M5, M9, and M10) compounds refer to Fig. 2. *: 4-Propanolguaiacol.

Response to Reviewer #3:

The manuscript entitled "Aqueous amine enables sustainable monosaccharide, monophenol and pyridine base coproduction in lignocellulosic biorefineries" by Qiu and coworkers describe a fractionation approach in which secondary aliphatic amine is used as reagent and base to produce monophenols and substituted pyridines.

The approach is rather novel and the authors should have credit for valorizing all parts of the lignocellulose. However, there are some weaknesses that need to be addressed by the authors.

1. First of all, lignin-first is not defined as lignin has a higher value than the other parts of the wood. The authors should look into the guidelines paper written a couple of years back to find an appropriate definition (this paper should be cited):

"lignin-first" biorefining approach, in which lignin valorisation is considered in the design phase.

Response: We appreciate your professional comments and apologize for providing an inappropriate definition of "lignin-first" biorefining. After referencing the article [*Abu-Omar, M. M. et al. Guidelines for performing lignin-first biorefining. Energy Environ. Sci., 2021, 14, 262*], we have revised the definition of "lignin-first" methods in the text and cited the reference. We would like to express our gratitude to the reviewer once again for providing valuable reference and suggestions.

Modification:

“Recently, recognizing the enormous potential of lignin as the most abundant aromatic reservoir in nature, a new biorefinery scheme has emerged, termed lignin-first, wherein the more efficient lignin-to-aromatic conversion is considered the priority over carbohydrate valorization. An elegant strategy has come into the spotlight, namely, the stabilization of active lignin species. α , β -diol group of β -O-4 motif was stabilized by adding formaldehyde to form 1,3-dioxane rings,¹⁰ stable acetal was formed between diols and unstable C2-aldehyde fragments,^{11, 12} and the C $_{\alpha}$ alcohol of β -O-4' was oxidized to a ketone.¹³” was changed to “Recently, recognizing the enormous potential of lignin as the most abundant aromatic reservoir in nature, a new biorefinery scheme has emerged, termed lignin-first, in which active stabilization approaches are developed to avoid condensation reactions that lead to more recalcitrant lignin structures.¹⁰ This lignin-first biorefining, which targets deriving more value from lignin than carbohydrate-first processing, is generally accomplished by the use of protection-group chemistries or reductive stabilization of reactive lignin intermediates.⁶ To name a few, α , γ -diol group of β -O-4' motif was stabilized by adding formaldehyde to form 1,3-dioxane rings,¹¹ stable acetal was formed between diols and unstable C2-aldehyde fragments,^{12, 13} the C $_{\alpha}$ alcohol of β -O-4' was oxidized to a ketone,¹⁴ and early-stage catalytic conversion of lignin was achieved by Reductive Catalytic Fractionation (RCF) with 2-propanol as an H-donor.^{15, 16}” in Lines 45–55 in the revised manuscript.

2. This has been demonstrated the last couple of years where lignin has been valorized and pulp has been upgraded to for instance dissolving pulp. Thus, it is not correct that carbohydrates generally are sacrificed.

Response: Thanks for your comments. We apologize for the unclear and absolute statements made in the previous version and have made revisions in the current version. We acknowledge the reviewer's point that carbohydrates are generally not sacrificed in lignin-first biorefining [*Adler, A. et al. Lignin-first biorefining of Nordic poplar to produce cellulose fibers could displace cotton production on agricultural lands. Joule 6, 1845–1858 (2022)*]. As you mentioned, carbohydrates can

be upgraded to dissolving pulp and/or can be hydrolyzed for sugar production. Our primary focus in this work is the enzymatic hydrolyzability of pretreated lignocellulose (cellulose and hemicellulose-rich fraction) to glucose and xylose. In lignin-first biorefining, when protective reagents such as formaldehyde and diols are used [Shuai, L. et al. Formaldehyde stabilization facilitates lignin monomer production during biomass depolymerization. *Science* 354, 329–333 (2016); Liu, Y. et al. Tunable and functional deep eutectic solvents for lignocellulose valorization. *Nat. Commun.*, 12, 5424 (2021); Lan, W. et al. Protection Group Effects During α,γ -Diol Lignin Stabilization Promote High-Selectivity Monomer Production. *Angew. Chem. Int. Ed.*, 57, 1356–1360 (2018).], or when RCF methods are employed [Ma, J. et al. Single-step conversion of wood lignin into phenolic amines. *Chem*, (2023); Sahayaraj, D. V. et al. An effective strategy to produce highly amenable cellulose and enhance lignin upgrading to aromatic and olefinic hydrocarbons. *Energy Environ. Sci.*, 16, 97–112 (2023).], a high cellulase dosage (30 FPU/g glucan or more) is often required to achieve high monosaccharide yields of substrates, as opposed to carbohydrate-first strategy where lower enzyme dosages (≤ 15 FPU/g glucan) suffice [Zhang, D.S. et al. Sulfite (SPORL) pretreatment of switchgrass for enzymatic saccharification. *Bioresour. Technol.*, 129, 127–134 (2013).]. The substrate pretreated by the DEA process in this study demonstrated outstanding enzymatic hydrolyzability, yielding 86% glucose and 78% xylose at a low cellulase dosage (8 FPU/g glucan).

Modification:

“On the downside, monomer production from the lignin portion in high yields was achieved at the expense of a reduction in the enzymatic hydrolyzability of pretreated lignocellulose. In other words, a high cellulase dosage (e.g., ~30 FPU/g glucan^{10, 12, 14}) is required to enable efficient enzymatic carbohydrate conversion.” was changed to “Although carbohydrates could be upgraded to dissolving pulp in lignin-first biorefining,¹⁷ the enzymatic hydrolyzability of pretreated lignocellulose did not perform as well as those obtained in carbohydrate-first strategy. In other words, a high cellulase dosage (~30 FPU/g glucan^{11, 13, 18} or more^{15, 19, 20}) is usually required to enable efficient enzymatic carbohydrate conversion.” in Lines 55–59 in the revised manuscript.

“Although lignin with protection groups prominently promoted lignin-derived monomer production, carbohydrate conversion in the lignin-first paradigm was impaired.” was changed to “Although lignin with protection groups prominently promoted lignin-derived monomer production, the lignocellulosic substrates from lignin-first biorefineries cannot be enzymatically hydrolyzed as effectively as those from carbohydrate-first strategies.” in Lines 409–412 in the revised manuscript.

3. When it comes to the results, it is important to understand what the 13 wt% yield monophenols correspond to. The authors should determine the beta-ether content of the feedstock.

Response: We greatly appreciate your review of our paper and the valuable feedback you have provided. The beta-ether content of cellulolytic enzyme lignin (CEL) isolated from the feedstock was determined to be 61 per 100 aromatic units, as confirmed by the 2D HSQC results (Fig. 3). In addition, the theoretical maximum monomer yields from CEL and DEA-L (lignin in the DEA pretreatment liquor) were estimated to be 34% and 18.5%, respectively, based on previously established methods [Lancefield, C. S. et al. *Isolation of Functionalized Phenolic Monomers through Selective Oxidation and C–O Bond Cleavage of the β -O-4 Linkages in Lignin*. *Angew. Chem. Int. Ed.*, 54, 258–262 (2015)]. The monophenol yields in our manuscript was from DEA-L not CEL, namely 15.6 wt% based on the lignin weight in the DEA pretreatment liquor (or 13wt% based on the lignin weight in untreated corn stover). Therefore, the depolymerization efficiency of DEA-L was up to 84% (15.6 wt% / 18.5 wt%).

Modification:

“The theoretical yield of monophenols from the lignins in the DEA and NaOH pretreatment liquors was estimated to be 18.5% and 13.2%, respectively, based on previously established methods.⁵¹ This suggested that the depolymerization efficiency of the DEA pretreatment liquor (84%) surpassed that of the NoOH pretreatment liquor (52%).” was added in Lines 254–257 in the revised manuscript.

Fig. 3 Structural comparison of cellulolytic enzyme lignin (CEL) from untreated CS, alkaline lignin (AL) from NaOH pretreatment, and DEA-treated lignin (DEA-L) from the fractionation with 40% v/v DEA. ^1H - ^{13}C cross signals and quantitative information on interunit linkages and lignin units.

4. When it comes to the N-containing aromatic compounds, the authors should determine the mass balance stemming from DEA and the biomass. How much DEA is incorporated into valuable products and how much can be recovered and re-used? DEA is missing in Fig 5.

Response: We appreciate your valuable suggestions. We concur with the reviewer's comments and have included a brief mass balance of DEA in Fig. 5b. DEA went to three categories: lost/unidentified (27%), consumed in amination of lignin and products (13%), and recycled (60%). Notably, the mass balance does not consider the DEA consumed in the reactions between

carbohydrates and DEA, given the complexity of these reactions. This portion should be relatively minor, as the loss of carbohydrates during pretreatment is limited.

Modification:

“In addition to the mass flow of lignocellulosic components, a mass balance of DEA was established, as presented in Fig. 5b. During the processing of CS, approximately 6% of DEA was lost due to its low boiling point. Another 1% and 2% of DEA were consumed in the amination of residual lignin and dissolved lignin, respectively. Approximately 91% of the unreacted DEA in the pretreatment liquor proceeded to the next step. In hydrogenolysis, 10% of DEA participated in the amination reactions, yielding N-functionalized monomers and oligomers. The remaining DEA could be recycled from the phenol and pyridine products via rectification, as confirmed by the Aspen simulation (Supplementary Figure 19).” was added in Lines 446–453 in the revised manuscript.

Fig. 5 Mass balance flow of polysaccharide (glucan and xylan) and lignin fractions of lignocellulose (a), and DEA (b). Pretreatment condition: 40% v/v DEA, 30 wt% biomass loading,

130 °C, and 1 h. Ctec2 loading used here is 15 FPU/g glucan.

5. The authors propose that the amine is substituting the benzylic alcohol, which sounds reasonable; it is not really understandable why the authors perform the hydrogenolysis not on the mixture but on certain models? What happens if product mixture (disclosed in Fig S18) would be used?

Response: We appreciate your professional feedback. The primary motivation behind our hydrogenolysis experiments using selected models (such as 4-hydroxyacetophenone, 4-hydroxy-3-methoxyphenylpyruvic acid, 4-hydroxyphenylacetic acid, and 4-hydroxy-3-methoxycinnamaldehyde) was to validate the hypothesis that pyridine bases could be generated through reactions between the oxidized side chains of lignin and amines. We intentionally selected model compounds bearing C=O groups, and these compounds were indeed detected in the hydrogenolysis products (Fig. 2) as well as in the β -O-4' model studies (Fig. S15). If product mixture from the β -O-4' model reactions was used, it would be difficult to elucidate the formation pathway of substituted pyridines because the mixed products shown in Fig. S18 (or Fig. S15 in the revised manuscript) are rather complex.

6. Furthermore, even hemicellulose could react with the amine, and this should be studied (furfural, 5-HMF or levulinic acid).

Response: We appreciate your valuable suggestion. First, we analyzed the DEA pretreatment liquor using HPLC and did not find furfural, 5-HMF, or levulinic acid, as shown in Supplementary Fig. 18. Second, we investigated the reactions between xylose and DEA using GC-MS (see Supplementary Note 4). GC-MS analysis of the products revealed the generation of cyclopentene derivatives, furan derivatives, and N-heterocyclic compounds.

Modification:

“Approximately 13% glucan and 8% xylan were not recovered during the DEA pretreatment. The common degradation products of carbohydrates reported in other works,^{59, 60} such as furfural, 5-hydroxymethylfurfural (5-HMF), and levulinic acid, were not detected in the DEA pretreatment liquor (Supplementary Fig. 18). To investigate potential side reactions between xylan and amines, we treated xylose in a 40% v/v DEA solution (Supplementary Note 4). The GC-MS analysis confirmed the presence of cyclopentene derivatives, furan derivatives, and N-heterocyclic compounds in the product mixture. Apart from producing small molecule compounds, brown nitrogenous polymers may be generated through Maillard reactions between carbohydrates and

amines.⁶¹” was added in Lines 436–445 in the revised manuscript.

Supplementary Figure 17. HPLC profiles of the DEA pretreatment liquor, levulinic acid, 5-HMF, and furfural.

The GC-MS result of the products from the reaction of xylose in 40% v/v DEA.

7. This reviewer does not agree that hydrogenolysis performed in water would have any advantages over organic solvents. Rather the contrary from an industrial point of view where HDO reactions are performed using carrier liquids to control reactivity.

Response: Thanks for your comments. We agree that water might not have any advantage in hydrogenolysis in some cases, compared with organic solvents, and the hydrogenolysis in organic solvents is more mature from an industrial perspective because most organic substrates and products have limited solubility in water. However, organic solvents are generally toxic, expensive, and unsafe compared with water. What we intended to state is that water is a safe and green solvent. If hydrogenolysis can be conducted in an aqueous medium with a similar or better performance to that in organic solvents, it is certainly of interest and significance to industry [Zhang, J. et al. *A Series of NiM (M = Ru, Rh, and Pd) Bimetallic Catalysts for Effective Lignin Hydrogenolysis in Water*. *ACS Catal.*, 4, 1574–1583 (2014); Qi, S. C. et al. *Catalytic hydrogenolysis of kraft lignin to monomers at high yield in alkaline water*. *Green Chem.* 19, 2636–2645 (2017).]. Our results indicate that the hydrogenolysis of DEA pretreatment liquor can be efficiently conducted without involving organic solvents. According to your comment, we have revised the related content in the revised manuscript to avoid confusion or misunderstanding.

Modification:

“Furthermore, simultaneous depolymerization and amination of lignin proceeded in an aqueous media by designing tandem amine-based fractionation/mild hydroprocessing strategy, thereby jointly producing monophenolics and pyridine bases.” was changed to “Furthermore, by designing a tandem amine-based fractionation/hydrogenolysis strategy, simultaneous depolymerization and amination of the dissolved lignins proceeded to co-produce monophenolics and pyridine bases.” in Lines 24–27 in the revised manuscript.

“Despite the above progress, the current protocol for N-participated lignin conversion still has some shortcomings, such as cumbersome steps, nonaqueous organic solvents used in the conversion, and low N-containing monomer yield.” was changed to “Despite the above progress, the current protocols for N-participated lignin conversion still have some shortcomings, such as cumbersome steps, and low N-containing monomer yield.” in Lines 234–236 in the revised manuscript.

Reviewers' Comments:

Reviewer #1:

Remarks to the Author:

The authors have spent good effort to offer an impressive revised version of the MS as well as reply letter. I am happy with the new content and recommend its acceptance in the current form.

Reviewer #2:

Remarks to the Author:

The authors describe an article entitled "Aqueous amine enables sustainable monosaccharide, monophenol, and pyridine base coproduction in lignocellulosic biorefineries". In this revised version, the manuscript has been improved by the authors according to the reviewers' comments. After inclusion of corrections and clarifications, this is now a manuscript that I recommend for publication.

Reviewer #3:

Remarks to the Author:

The revised manuscript by Lou, Qiu and coworkers entitled "Aqueous amine enables sustainable monosaccharide, monophenol, and pyridine base coproduction in lignocellulosic biorefineries" is an improved version of the previous submission where many of the questions that this reviewer raised were answered.

However, there are two questions that still remain unanswered in this reviewer's opinion that makes the manuscript difficult to assess for future readers.

1) This reviewer acknowledges the attempt to determine the theoretical maximum yield of monophenols. However, would appreciate if the authors could use any of the protocols suggested in the guidelines paper, NBO or thioacidolysis?

Fig 3. HSQC should not be written as quantitative....

2) this reviewer is still confused on the mass balances, but thank the authors for providing the additional Sankey. This reviewer screened material in the main manuscript and SI and could still not get a full picture on how the experiment was performed: and the resulting mass balances. Now, it is read as 30% of 40% DEA is lost during the reaction which equals to 12 wt% of initial loading and that 30% of the 74% of lignin content which is (16% of original biomass) is converted to monomers inc N-containing, and this should equal to 3.5%.

In the current manuscript, this is very difficult to decipher. Could the authors please provide this data. In addition, the authors should report their experiments using a conservative approach: ml/g (mmol, when applicable) and not the 30 v/v.

Manuscript ID: NCOMMS-23-32549A

Manuscript Title: Aqueous amine enables sustainable monosaccharide, monophenol, and pyridine base coproduction in lignocellulosic biorefineries

Response to Reviewer #1:

The authors have spent good effort to offer an impressive revised version of the MS as well as reply letter. I am happy with the new content and recommend its acceptance in the current form.

Response: Thanks to your interest in this manuscript. We thank the reviewer for the time and effort that have put into reviewing our manuscript.

Response to Reviewer #2:

The authors describe an article entitled "Aqueous amine enables sustainable monosaccharide, monophenol, and pyridine base coproduction in lignocellulosic biorefineries". In this revised version, the manuscript has been improved by the authors according to the reviewers' comments. After inclusion of corrections and clarifications, this is now a manuscript that I recommend for publication.

Response: Thank you for your suggestions in improving the previous version of manuscript. We thank the reviewer for the time and effort that have put into reviewing our manuscript.

Response to Reviewer #3:

The revised manuscript by Lou, Qiu and coworkers entitled "Aqueous amine enables sustainable monosaccharide, monophenol, and pyridine base coproduction in lignocellulosic biorefineries" is an improved version of the previous submission where many of the questions that this reviewer raised were answered.

However, there are two questions that still remain unanswered in this reviewer's opinion that makes the manuscript difficult to assess for future readers.

1. This reviewer acknowledges the attempt to determine the theoretical maximum yield of monophenols. However, would appreciate if the authors could use any of the protocols suggested in the guidelines paper, NBO or thioacidolysis?

Response: Thanks to your valuable suggestions. According to the previously established

protocols [Yamamura, M. et al. *Microscale alkaline nitrobenzene oxidation method for high-throughput determination of lignin aromatic components. Plant Biotechnol.* 27, 305-310 (2010).], nitrobenzene oxidation (NBO) was performed to determine the theoretical maximum yield of monophenols of DEA-L and AL, and the corresponding values were 27.5% and 21.0%, respectively.

Modification:

“Nitrobenzene oxidation method: Nitrobenzene oxidation of DEA-L and AL were conducted following previously established procedures.⁷ Briefly, 40 mg of dried lignin was dissolved in a mixture of 2 N NaOH (4 mL) and nitrobenzene (0.24 mL) in a Teflon-lined reactor and heated in an oven at 170 °C for 1 h. The reaction mixture was centrifuged and then extracted with ethyl acetate (30 mL × 3) to remove unreacted nitrobenzene. The alkaline water layer was acidified to pH 2 with a 2 N HCl solution and extracted with ethyl acetate (30 mL × 3). The organic layer was washed with brine and dried over anhydrous Na₂SO₄, and the solvent was removed by rotary evaporation. The resultant products were dissolved in 2 mL of acetone with decane as an IS. Silylations of the sample solution were performed with BSTFA prior to GC-MS analyses.” was added in Page 8 in SI.

$$\text{Theoretical maximum yield (\%)} = \frac{m_{\text{monophenols}} \text{ (mg)}}{40 \text{ mg} \times \text{lignin purity (\%)}} \quad (\text{S9})$$

“The theoretical yield of monophenols from the lignins in the DEA and NaOH pretreatment liquors was estimated to be 18.5% and 13.2%, respectively, based on previously established methods.⁵¹ This suggested that the depolymerization efficiency of the DEA pretreatment liquor (84%) surpassed that of the NaOH pretreatment liquor (52%).” was changed to “The theoretical maximum yield of monophenols for the lignins isolated from the DEA and NaOH pretreatment liquors was estimated to be 27.5% and 21.0%, respectively, using the nitrobenzene oxidation method. This suggested that the depolymerization efficiency of the lignins in the DEA pretreatment liquors (57%) surpassed that of the ones in the NaOH pretreatment liquors (32%).” in Lines 261–265 in the revised manuscript.

2. Fig 3. HSQC should not be written as quantitative...

Response: Thanks for your careful checks. We are sorry for the lack of rigor in the previous version on the caption of Fig. 3. According to the reference [Phongpreecha, T. et al. *Predicting lignin depolymerization yields from quantifiable properties using fractionated biorefinery lignins. Green Chem.*, 2017, 19, 5131], the HSQC technology has been expressed as semi-quantitative in the revised

manuscript.

Modification:

“ ^1H - ^{13}C cross signals and quantitative information on interunit linkages and lignin units.” was changed to “ ^1H - ^{13}C cross signals and semi-quantitative information on interunit linkages and lignin units.” in Lines 337–338 in the revised manuscript.

3. this reviewer is still confused on the mass balances, but thank the authors for providing the additional Sankey. This reviewer screened material in the main manuscript and SI and could still not get a full picture on how the experiment was performed: and the resulting mass balances. Now, it is read as 30% of 40% DEA is lost during the reaction which equals to 12 wt% of initial loading and that 30% of the 74% of lignin content which is (16% of original biomass) is converted to monomers inc N-containing, and this should equal to 3.5%. In the current manuscript, this is very difficult to decipher. Could the authors please provide this data.

Response: Thanks to your professional comments. We acknowledged that the previous version of the mass balance was hard to decipher, and some experimental details were missing. One thing we want to explain to the reviewer is that each stream in the mass balance is based on the initial loading. For instance, the yield of phenolic and N-containing monomers (30%) is based on the weight of lignin in raw materials rather than the soluble lignin. In addition, the mass balance began from the initial DEA loading not the 40% DEA. In order to resolve these confusing problems, the methodologies used to figure out the mass balances of DEA were added in SI (see Page 9), and the weight of each stream for lignocellulosic components and DEA was added in Fig. 5 on a basis of 100 g of biomass.

Modification:

“*Mass balance of DEA:* The DEA consumed in residual and dissolved lignin was calculated based on the N content of DEA-treated corn stover (N: 0.23%) and DEA-L (N: 3%), respectively. The loss of 40% DEA was assessed by recording the weight changes in the overall material before and after pretreatments. To figure out DEA loss in hydrogenolysis, HPLC equipped with an RID was employed to analyze the DEA content variations in a 4% DEA solution under hydrogenolysis conditions. For the determination of DEA involved in the amination of hydrogenolysis products (including all aminated products), the dried hydrogenolysis products obtained through rotary evaporation and subsequent freeze-drying were subjected to an N elemental analysis (N: 5.2%).” was added in Page 9 in SI.

Fig. 5 Mass balance flow of polysaccharide (glucan and xylan), lignin fractions, and DEA on a basis of 100 g of biomass. Pretreatment condition: 40% v/v DEA, 30 wt% biomass loading, 130 °C, and 1 h. CTec2 loading used here is 15 FPU/g glucan. *: All aminated products in the hydrogenolysis liquors.

4. In addition, the authors should report their experiments using a conservative approach: ml/g (mmol, when applicable) and not the 30 v/v.

Response: We greatly appreciate this insightful suggestion. We agree that the expression of DEA concentration, mL/g, is more scientific than the one (v/v), and the conservative approach has been added in the revised manuscript accordingly. On the other hand, the previous approach (v/v) was also remained because it led to a better understanding for readers how did we prepare the aqueous solution of DEA.

Modification:

“In DEA-based fractionation, the lignin removal was low (31.8% or 20.0%) under the condition without water (pure DEA) or with too much water (5% v/v DEA), while excellent

delignification was observed at a DEA content in the range of 20% to 80%. Pretreatments with 60% and 40% DEA resulted in a similar lignin removal (~74%), which outperformed ammonia and DES.” was changed to “In DEA-based fractionation, the lignin removal was low (31.8% or 20.0%) under the condition without water (1.41 mL DEA/g, namely pure DEA) or with too much water (0.05 mL DEA/g, namely DEA: water = 5: 95 v/v), while excellent delignification was observed at a DEA content in the range of 0.21 mL/g (DEA: water = 20: 80 v/v) to 1.22 mL/g (DEA: water = 80: 20 v/v). Pretreatments with 0.79 (DEA: water = 60: 40 v/v) and 0.48 mL DEA/g (DEA: water = 40: 60 v/v) resulted in a similar lignin removal (~74%), which outperformed ammonia and DES.” in Lines 119–123 in the revised manuscript.

Reviewers' Comments:

Reviewer #3:

Remarks to the Author:

The authors have taken the opportunity to revise their manuscript according to suggestions given by this reviewer.

This reviewer thinks that the manuscript can be accepted for publication in Nature communications.